# *Lactobacillus fermentum* Alleviates the Colorectal Inflammation Induced by Low-Dose Sub-Chronic Microcystin-LR Exposure

**DOI:** 10.3390/toxins15090579

**Published:** 2023-09-19

**Authors:** Yue Yang, Cong Wen, Shuilin Zheng, Fengmei Song, Ying Liu, Xueqiong Yao, Yan Tang, Xiangling Feng, Jihua Chen, Fei Yang

**Affiliations:** 1The Key Laboratory of Typical Environmental Pollution and Health Hazards of Hunan Province, Department of Epidemiology and Health Statistics, School of Public Health, Hengyang Medical School, University of South China, Hengyang 421001, China; yangy930806@126.com (Y.Y.); songfengmei2020@163.com (F.S.); liuying@usc.edu.cn (Y.L.); yxq1212@usc.edu.cn (X.Y.); jiayi1530@163.com (Y.T.); 2Hunan Provincial Key Laboratory of Clinical Epidemiology, Xiangya School of Public Health, Central South University, Changsha 410017, China; fengxl2011@csu.edu.cn (X.F.); chenjh@csu.edu.cn (J.C.); 3Changsha Yuhua District Center for Disease Control and Prevention, Changsha 410014, China; 15874259745@163.com; 4Changsha Center for Disease Control and Prevention, Changsha 410004, China; zhengshuilin95@163.com; 5Hengyang Medical School, The First Affiliated Hospital, University of South China, Hengyang 421001, China

**Keywords:** microcystin-LR, probiotics, colorectal inflammation, intestinal diseases, *Lactobacillus fermentum*

## Abstract

Microcystin-LR (MC-LR) contamination is a worldwide environmental problem that poses a grave threat to the water ecosystem and public health. Exposure to MC-LR has been associated with the development of intestinal injury, but there are no effective treatments for MC-LR-induced intestinal disease. Probiotics are “live microorganisms that are beneficial to the health of the host when administered in sufficient quantities”. It has been demonstrated that probiotics can prevent or treat a variety of human diseases; however, their ability to mitigate MC-LR-induced intestinal harm has not yet been investigated. The objective of this study was to determine whether probiotics can mitigate MC-LR-induced intestinal toxicity and its underlying mechanisms. We first evaluated the pathological changes in colorectal tissues using an animal model with sub-chronic exposure to low-dose MC-LR, HE staining to assess colorectal histopathologic changes, qPCR to detect the expression levels of inflammatory factors in colorectal tissues, and WB to detect the alterations on CSF1R signaling pathway proteins in colorectal tissues. Microbial sequencing analysis and screening of fecal microorganisms differential to MC-LR treatment in mice. To investigate the role of microorganisms in MC-LR-induced colorectal injury, an in vitro model of MC-LR co-treatment with microorganisms was developed. Our findings demonstrated that MC-LR treatment induced an inflammatory response in mouse colorectal tissues, promoted the expression of inflammatory factors, activated the CSF1R signaling pathway, and significantly decreased the abundance of *Lactobacillus*. In a model of co-treatment with MC-LR and *Lactobacillus fermentum* (*L. fermentum*), it was discovered that *L. fermentum* substantially reduced the incidence of the colorectal inflammatory response induced by MC-LR and inhibited the protein expression of the CSF1R signaling pathway. This is the first study to suggest that *L. fermentum* inhibits the CSF1R signaling pathway to reduce the incidence of MC-LR-induced colorectal inflammation. This research may provide an excellent experimental foundation for the development of strategies for the prevention and treatment of intestinal diseases in MC-LR.

## 1. Introduction

In recent years, the eutrophication of water bodies with nitrogen and phosphorus and the intensification of the global warming trend have led to frequent outbreaks of cyanobacterial blooms in many regions of the world, and the release of a variety of highly active and toxic cyanobacterial toxins is one of the main dangers associated with blooms [1,2]. Cyanobacteria and cyanobacterial toxin pollution has been a significant environmental issue in China and the rest of the world for quite some time, with Microcystin-LR (MC-LR) being the cyanobacterial toxin with the widest distribution, highest production, and most severe toxicity [3,4,5]. In addition to being detected in natural water bodies such as rivers and lakes, MC-LR has also been detected in water factory water, terminal water, and bottled/barrel water. Furthermore, MC-LR can be enriched in a variety of agricultural and aquatic products, making the current pollution situation extremely grave [6,7]. Extremely hazardous MC-LR can be ingested through drinking water and aquatic products, causing injury to multiple organs, including the liver, kidneys, intestines, nerves, immune system, and reproductive organs [8,9]. MCs typically enter the human body via the oral route, such as drinking water and food, and the gastrointestinal tract is the first organ to ingest and be exposed to MCs [8]. Using a retrospective cohort study, Zhou et al. discovered that the level of exposure to MC-LR was positively correlated with the incidence of colorectal cancer [10]. Sedan et al. found that MC-LR significantly decreased intraepithelial lymphocytes in the intestine [11]. Low doses of MC-LR induced severe inflammatory responses and fibrogenesis in mouse intestines, according to our previous research [8].

There are 100 trillion microorganisms (including bacteria, viruses, fungi, and protists) in the human gut. These microorganisms can attach to the epithelium or mucus layer and produce biofilms that have a significant impact on host development [12]. Alterations in the intestinal microbiota are strongly linked to a wide range of diseases, particularly intestinal disorders. In a 96 h study conducted by Zhang et al., adult male crayfish were exposed to varying concentrations of MC-LR. After exposure to MC-LR, the abundance of bacteria such as *Firmicutes* and *Planctomycetes* at the phylum level and bacteria such as *Dysgonomonas*, *Brevundimonas*, and *Anaerorhabdus* at the genus level were significantly altered in crayfish intestines. In crayfish, the elimination of gut microorganisms can cause metabolic and genetic abnormalities related to gut microorganisms [13]. Li et al. discovered that exposure of African clawed toads to MC-LR reduced the abundance of *Firmicutes* and *Bacteroidetes* in their intestinal tracts, while Fusobacterium was significantly enriched [14]. Ding et al. discovered that MC-LR treatment decreased the relative abundance of zebrafish Vibrio bacteria while increasing the relative abundance of *Methylobacterium* and *Pseudocardia* bacteria [15]. MC-LR affects the diversity and abundance of intestinal flora, leading to intestinal microecological dysregulation, impaired nutrient absorption and metabolism, the inhibition of intestinal probiotic growth, the promotion of pathogenic bacterial colonization, and the induction of host intestinal diseases. No study has investigated the role of probiotics in MC-LR-induced intestinal injury.

Probiotics are “live microorganisms that are beneficial to the health of the host when administered in sufficient quantities” [16]. These probiotics are used to treat a variety of conditions, such as constipation, diarrhea, irritable bowel syndrome, allergies, obesity, diabetes, and cancer, among others [17]. Numerous studies have shown that probiotics play a crucial role in preventing or treating intestinal diseases. Several studies have demonstrated that probiotics can enhance the intestinal mucosal auto-preventive mechanism, antagonize the pathogenic effects of pathogenic bacteria, regulate the release of inflammatory factors, and quickly restore the intestinal flora [18]. *Clostridium difficile spp.* affects the accumulation of intraepithelial lymphocytes (IELs) and increases the number of regulatory T cells (Treg cells) in the colon, which downregulates the expression of inflammatory cytokines (TNF-, IL-12, IFN-, IL-1, and IL-6) and upregulates the expression of inhibitory cytokines (IL-10). Through inhibiting histone deacetylase (HDAC), butyrate produced by Clostridium butyricum can directly promote the differentiation of Tregs [19,20]. Recent evidence suggests that probiotics can influence immune regulation through modulating dendritic cell (DC) maturation and the production of tolerogenic DCs (tolDCs), which may then inhibit inflammatory responses. PSA from outer membrane vesicles (OMVs) derived from Bifidobacterium fragilis ameliorates DSS-induced colitis in rodents through affecting DCs [21]. MUC-2 levels are positively correlated with *Akkermansia* abundance in rodents with DSS-induced colitis, and *Akkermansia* supplementation can strengthen the colonic mucus barrier through increasing the thickness of the colonic mucus layer [22]. In conclusion, probiotics and their metabolites can modulate intestinal immune function and intestinal permeability as well as reduce intestinal inflammatory response via multiple mechanisms. Using an animal model, Zhao et al. demonstrated for the first time that a probiotic mixed flora (containing, among others, *Lactobacillus rhamnosus* and *Bifidobacterium bifidum*) ameliorated the hepatotoxicity of MC-LR and acted as an antidote [23]. To date, no research has examined whether probiotics have the same protective effect against MC-LR-induced intestinal toxicity.

CSF1R is a transmembrane tyrosine/serine kinase receptor expressed normally in mononuclear phagocytes. CSF1R is widely believed to be activated by autocrine or paracrine mechanisms and to play a central role in numerous diseases, including chronic inflammatory disorders, bone diseases, and tumors [24,25]. Long-term exposure to low-dose MC-LR activated the CSF1R/Rap1b signaling pathway through inducing oxidative stress and promoted the release of inflammatory and fibrosis-related factors, which led to the development of chronic inflammation and fibrosis in the colorectal tissue [8]. The CSF1R signaling pathway plays a crucial role in intestinal injury induced by MC-LR. Therefore, in this study, we also investigated the expression of proteins associated with the CSF1R signaling pathway to determine if the microorganisms we screened could influence MC-LR-induced colorectal injury via this crucial signaling pathway.

In this investigation, mice were exposed to low-dose MC-LR for six months in order to analyze and screen mouse feces for different microorganisms using microbial sequencing. And for the first time, the mitigating effect of microorganisms (particularly probiotics) on colorectal injury induced by low-dose sub-chronic MC-LR exposure was investigated using an in vitro model of microorganism and MC-LR co-treatment. This research may provide a solid theoretical and laboratory foundation for the development of preventative and therapeutic strategies for MC-LR-induced intestinal diseases.

## 2. Results

### 2.1. The Body Weight of MC-LR-Induced Mice

Throughout the duration of exposure, body weight was measured every two weeks. Neither mouse mortality nor abnormal behavior was observed during the 6-month investigation. As shown in Figure 1, the MC-LR group had no effect on the weight change of mice.

### 2.2. Histopathology Damage of MC-LR-Induced Mice

Each mouse’s colorectal tissue was stained with Hematoxylin-Eosin (HE), and Figure 2 depicts the pathological alterations of colorectal tissue. MC-LR can induce lymphocyte infiltration and inflammatory response in colorectal tissues relative to the control (CT) group.

### 2.3. The Expression of Inflammatory Factors and CSF1R/Rap1b Signaling Pathway-Related Proteins in MC-LR-Induced Mice

In order to further verify the occurrence of colorectal tissue inflammation caused by MC-LR-treatment, we detected the expression level of inflammatory factors in colorectal tissue of mice via qPCR. The results are illustrated in Figure 3. In the MC-LR treatment groups, the expression of inflammatory factors (IL-6, TNF-α, and IL-1β) was increased relative to the CT group.

In our previous studies, it was found that MC-LR can induce chronic inflammation in colorectal through activating CSF1R/Rap1b signaling pathway [8]. Therefore, in this study, we verified the changes in the expression levels of CSF1R and Rap1b. Figure 4 demonstrates the outcomes. The expression of CSF1R and Rap1b were significantly up-regulated in the MC-LR + HFD treatment groups compared to the CT group.

### 2.4. Gut Microbiome Alterations of MC-LR-Induced Mice

*Bacteroidetes* and *Firmicutes* were the most represented groups in terms of phylum classification (Figure 5A). The twenty most common genera accounted for approximately 20% of all microorganisms (Figure 5B). To investigate the specific bacterial taxa associated with MC-LR exposure, the greatest taxonomic differences at the genus level were represented by the LDA score (Figure 5C). At the genus level, the *Lactobacillus* decreased in the MC-LR treatment groups (Figure 5D).

### 2.5. Effect of Lactobacillus fermentum (L. fermentum) Treatment on the Expression of CSF1R/Rap1b Signaling Pathway-Related Proteins and the Inflammatory Factors in MC-LR-Treated Cell Model

In addition, using normal colonic epithelial NCM460 cells, we conducted a preliminary test to determine whether *L. fermentum* can ameliorate the colorectal damage caused by MC-LR in an in vitro model. qPCR analysis revealed a significant upregulation of CSF1R/Rap1b and inflammatory factors (IL-6, TNF-α, and IL-1β) in the MC-LR-treated group (Figure 6). In contrast, after *L. fermentum* treatment, these expression level changes were substantially attenuated (Figure 7).

## 3. Discussion

In our dyeing experiments conducted on mice with sub-chronic exposure to low doses of MC-LR, we observed no significant difference in the trend of body weight gain between the MC-LR-treated group and the control group, indicating that sub-chronic exposure to low doses of MC-LR has no effect on mouse body weight. Consistent with our findings, Su et al. found that MC-LR gavage at 1000 g/kg for seven days had no effect on body weight changes in rodents compared to controls [26,27]. Similarly, Pan et al. found that mice exposed to varying concentrations of MC-LR for 90 days did not experience a significant decrease in body weight [28]. The failure of long-term exposure to low-dose MC-LR to cause weight loss in mice may be due to the fact that the dose was low and had not yet reached the threshold that could lead to changes in energy metabolism and other effects, or that low-dose stimulation and activation of the protective mechanism of the mice resulted in a tolerance. In addition, we evaluated the histopathological changes in the colorectum and found that mice with sub-chronic exposure to low-dose MC-LR had substantially increased colorectal inflammatory cell infiltration and significant changes in inflammatory response. He et al. demonstrated that MC-LR-exposed intestinal tissues exhibited a highly disorganized epithelial cell configuration, the absence of villi, and a significant number of eosinophils [29]. Zhang et al. also discovered that MC-LR-treated groups at 10 or 40 μg/L exhibited significant eosinophilic granulocytosis, lymphocytic infiltration, and epithelial vacuolization [30].

Sub-chronic exposure to modest doses of MC-LR increased the expression of inflammation-related genes in the colorectal tissues of mice, corroborating our previous report on the inflammatory-responsive effects of MC-LR on the small intestine [31]. The expression of pro-inflammatory factors such as IL-6, TNF-, and IL-1 was substantially higher in the MC-LR-treated group compared to the CT group. Similar to our findings, Li et al. found substantially elevated expression levels of TNF- and IL-1 in African Xenopus gut tissues treated with MC-LR [32]. According to the findings of Pan et al., mice exposed to varied concentrations of MC-LR for 90 days exhibited a dose-dependent increase in the expression of IL-6 and TNF- in tissues [29]. Ding et al. demonstrated that tadpoles exposed to MC-LR for 30 days had substantially elevated levels of IL-1 expression in their intestinal tissues, indicating the occurrence of significant inflammatory responses [15]. Similarly, Duan et al. found that the relative expression of the pro-inflammatory factors MyD88, Rel, and TNF-a in the intestinal tissues of crustaceans *Penaeus vannamei* exposed to MC-LR for 72 h was significantly increased and triggered the body’s immune response [33]. Our findings demonstrate for the first time that sub-chronic exposure to modest doses of MC-LR can result in tissue damage and inflammatory responses in mice with colorectal tissue.

Given the importance of gut microorganisms in all aspects of human health and the severe toxicity of environmental pollutants to the body, it is necessary to investigate the impact of changes in the composition of gut microorganisms caused by sub-chronic exposure to low-dose MC-LR [34,35]. Sub-chronic exposure to modest doses of MC-LR led to significant changes in the composition of the intestinal flora, according to the findings of this study. *Lactobacillus* levels in the feces of rodents treated with MC-LR were drastically reduced compared to the control group. *Lactobacillus* are rod-shaped, Gram-positive, partially anaerobic or mildly aerobic, lactic acid-producing bacteria [36]. *Lactobacillus* is a probiotic that contributes significantly to human health [37]. *Lactobacillus* can inhibit the development of harmful bacteria, maintain intestinal flora balance, and prevent intestinal infections and inflammation [38]. *Lactobacillus* degrades fiber and other indigestible food components, thereby facilitating digestion and nutrient absorption [39]. *Lactobacillus* reinforces the intestinal mucosal barrier, preventing harmful substances from accessing the bloodstream and enhancing immunity [36]. *Lactobacillus* is also capable of regulating immune system response and reducing allergic reaction symptoms [36]. Therefore, we hypothesized that Lactobacillus bacteria might protect against MC-LR-induced intestinal injury.

We constructed a cell model of MC-LR co-treated with *Lactobacillus* to determine whether *Lactobacillus* could reduce the intestinal inflammatory response caused by MC-LR. Our results showed that supplementation with *L. fermentum* significantly alleviated MC-LR induced colorectal inflammation. In line with our findings, numerous studies have demonstrated that *L. fermentum* can mitigate intestinal damage induced by various factors. Jian et al. demonstrated that fecal microbiota transplantation (FMT) of *Lactobacillus plantarum* to normal mice or elimination of *Lactobacillus* in the mouse gut using antibiotics significantly reduced irradiation-induced intestinal damage and averted mouse mortality [40]. According to Zhou et al., the administration of *Lactobacillus plantarum* substantially increased the number of intestinal stem cells (ISCs) and the expression of differentiation markers of cuprocytes, enterocytes, and enteroendocrine cells in the mouse colon. *Lactobacillus plantarum* stimulates the regeneration of colonic crypts in mice with colitis through regulating the proliferation and differentiation of intestinal stem cells (ISCs) and through modifying the gut microbiota profile [41]. Liu et al. found that *Lactobacillus plantarum* 23-1 significantly reduced body mass and adiposity index, improved serum and hepatic lipid levels, attenuated histopathological damage in the liver and small intestine, attenuated obesity-induced oxidative stress and inflammatory responses, and alleviated symptoms of obesity in HFD-fed mice [42]. Li et al. also discovered that *Lactobacillus plantarum* J26 could maintain the intestinal barrier, prevent LPS from crossing the intestinal barrier, correct the disorders of the intestinal-hepatic axis, inhibit the activation of the Toll-like receptor 4 (TLR4)-mediated MAPK signaling pathway, reduce hepatic inflammation, and restore liver function [43].

Our previous study showed that long-term exposure to low-dose MC-LR increased chronic colorectal inflammation, fibrosis, and barrier damage through activating the CSF1R/Rap1b signaling pathway [8]. Therefore, in this paper, we also verified whether *L. fermentum* inhibits CSF1R signaling pathway activation induced by MC-LR. The MC-LR+*L. fermentum* treatment group significantly inhibited the activation of the CSF1R signaling pathway induced by MC-LR treatment, indicating that *L. fermentum* can inhibit the CSF1R signaling pathway. Through inhibiting the CSF1R signaling pathway, our findings demonstrate for the first time that *L. fermentum* may exert anti-inflammatory effects to mitigate MC-LR-induced intestinal toxicity. Several extra studies have investigated the ameliorative effects of probiotics on injury-induced gut microbiota disorders. Wang et al. demonstrated that *Lactobacillus plantarum* DP189 remodeled the gut microbiota and reduced neurodegenerative lesions in rodents with Parkinson’s disease through reducing the number of pathogenic bacteria and increasing the abundance of probiotic bacteria. However, the present study has not yet investigated the mitigating effect of *Lactobacillus* on the MC-LR-induced disruption of intestinal microbes. Future in-depth studies may look into the role and mechanism of intestinal microbial disruption and remodeling in MC-LR-induced intestinal injury. Despite the fact that numerous studies have been conducted on the degradation of environmental pollutants by microorganisms [4], the present study has not yet determined whether the mitigating effect of *L. fermentum* on the enterotoxicity of MC-LR is due to its inhibitory effect on the release of inflammatory factors or its possible degrading effect on MC-LR. Therefore, the specific role and mechanism of *L. fermentum* or other probiotics in mitigating the damage caused by environmental pollutants can be further investigated, providing a solid foundation for the development of future disease treatment strategies and the search for specialized antidotes.

## 4. Conclusions

In this study, we discovered that sub-chronic exposure to low-dose MC-LR caused rodents to develop colorectal damage and an inflammatory response. It reveals for the first time the mitigating effect of *L. fermentum* on MC-LR-induced colorectal injury in mice, as well as the plausible mechanisms underlying this effect. The present study can establish the theoretical groundwork and provide new technical support for the development of safe and effective biocontrol technology to combat algal toxin pollution, as well as the creation of effective antidotes to mitigate the harmful effects of MC-LR.

## 5. Materials and Methods

### 5.1. Chemicals and Reagents

MC-LR with a purity ≥ 95% was purchased from Alexis Corporation (Lausen, Switzerland). *L. fermentum* was acquired from Henan BeNa Culture collection (BNCC). Normal colonic epithelial NCM460 cells were purchased from Otwo Biotech Inc., (Shenzhen, China). RPMI 1640 medium was purchased from Sigma (Livonia, MI, USA). RIPA buffer, bovine serum albumin (BSA), and the Pierce bicinchoninic acid (BCA) protein assay kit were purchased from Thermo Scientific Pierce (Rockford, IL, USA). The polyvinylidene fluoride (PVDF) membrane was purchased from Merck Millipore Ltd., (Billerica, MA, USA). Trizol Reagent was purchased from TaKaRa LA Taq (Shiga, Japan). The ChamQ Universal SYBR qRT-PCR Master Mix and HiScript^®^ II Q RT SuperMix for qRT-PCR were purchased from Vazyme (Nanjing, China). The IL-6, TNF-α, IL-1β, and IL-10 ELISA kits were purchased from Neobioscience (Shenzhen, China). CSF1R antibody and Rap1b antibody were purchased from Abcam (Cambridge, UK). β-actin antibody and HRP-conjugated Affinipure Goat Anti-Rabbit IgG(H+L) were purchased from Proteintech (Wuhan, China).

### 5.2. Design of Animal Experiment

Six-to-eight-week-old, 20–22 g male C57BL/6J SPF mice were obtained from Hunan SJA Laboratory Animal Co., Ltd. and deposited at the Experimental Animal Centre of Central South University, China. The animals were fed standard rodent pellets and kept on a light–dark cycle of 12:12 h. Forty mice were randomly assigned to four groups: 1, 60, and 120 g/L MC-LR and a CT group. In this study, the MC-LR treatment concentration was selected according to WHO guidelines, peer literature, and our previous published papers [8,44,45,46]. According to the results of Fawell et al. and Zhao et al., concentrations of 1/2 NOAEL (120 µg/L) and 1/4 NOAEL (60 µg/L) of MC-LR were selected in this study as treatment doses for long-term exposure to low-dose MC-LR [45,46]. In order to investigate whether the maximum allowable concentration of MC-LR for humans would also cause injury to animals, we have added a 1 μg/L MC-LR treatment group according to WHO guidelines [44]. To assure statistical validity, our experimental design allocated 10 mice per treatment group to eliminate individual animal differences. And in accordance with the initiative of animal protection organizations and animal welfare, we followed the 4R principle to reduce the number of rodents utilized through sharing the grouping with other experimental designs. MC-LR was administered orally (via ingesting water) to mice in each group for six months at the concentrations indicated. Once every two weeks, body weight was determined. All experimental protocols were authorized by the Central South University Animal Care and Use Committee (permit number XYGW-2018-41). Serum and colorectal tissues were collected 24 h after the last day of the experiment.

### 5.3. Histological Analysis

Samples of colorectal tissue were immediately isolated and evaluated. The bloodstains were washed away with icy phosphate-buffered solution (PBS, pH = 7.2) and, afterwards, kept at room temperature overnight with 4% paraformaldehyde (PFA) within a sterile PBS buffer. The tissues had been encased in paraffin. After waxing, 4 micrometer-thick tissue splits were stained with HE. Using an optical microscope (Motic (Xiamen, China), BA210), the sections were then observed and photographed.

### 5.4. Metagenomic Sequencing

Wuhan Comtest Technology Co., Ltd. (Wuhan, China) was entrusted with the sequencing of mouse fecal microorganisms. Typically, the target sequences reflecting the composition and diversity of microbial colonies, such as microbial ribosomal RNA, are used as targets, and the corresponding primers are designed based on the conserved regions of the sequences, and sample-specific Barcode sequences are added, so as to amplify the variable regions of the rRNA genes or the specific gene fragments via PCR. PCR amplification is performed using the NEB Company’s Q5 high-fidelity DNA polymerase, and the amplification cycle is rigorously regulated. This endeavor utilized the Illumina MiSeq platform for the paired-end sequencing of community DNA fragments. The obtained sequences were classified into operational taxonomic units (OTUs), the diversity level of each sample was determined based on the abundance distribution of OTUs across samples, and the specific composition of each sample was analyzed at various taxonomic levels. The procedures were precisely as described by Yang et al. [8]. By means of LEfSe analysis, key differential microorganisms were screened for.

### 5.5. Design of Cell Experiment

We established an in vitro model of MC-LR with or without *L. fermentum* exposure. NCM460 normal colonic epithelial cells (Otwo Biotech Inc., Shenzhen, China) were maintained in RPMI 1640 medium supplemented with 10% FBS and tetracycline in a 5% CO_2_ cell incubator. Cells were separated into the following groups: CT; MC-LR; *L. fermentum*; MC-LR+*L. fermentum*. MC-LR were briefly rinsed twice with PBS and reconstituted in RPMI 1640. In six-well tissue culture plates, 1 × 10^6^ NCM460 cells were seeded and allowed to adhere overnight under incubation conditions. When the degree of cell confluence reached 60–70%, the cells were washed twice using PBS, and the serum-free medium was replaced to continue culturing the cells for 24 h to keep the cells in the same cell cycle. Then, in each well, either RPMI 1640 or 10^8^ CFU/mL *L. fermentum* were added. In cellular experiments, we calculated the treatment dose of MC-LR for cellular experiments based on WHO guidelines and the MC-LR treatment dose from the acute research experiments of Fawell et al., in conjunction with the formula for converting the doses for cellular and animal experiments [44,45]. The final concentration of MC-LR was 200 μmol/L after dilution with RPMI 1640. After 1 h of incubation at 37 °C in 5% CO_2_, the supernatant was discarded [47,48]. After washing the cells with phosphate-buffered solution, total RNA and protein were extracted from each cohort.

### 5.6. Western Blotting (WB)

Proteins were isolated using sodium dodecyl sulphate-polyacrylamide gel electrophoresis (SDS-PAGE) and electroblotted onto a PVDF membrane. Protein Free Rapid Blocking Buffer was utilized to inhibit the membrane transfer. Antibodies were incubated with membranes overnight at 4 °C. The transplanted membrane was incubated for 60 m with goat anti-mouse IgG (H+L) HRP conjugate or goat anti-rabbit IgG (H+L) HRP conjugate. Using a Bio-Rad chemiluminescence imaging system (Bio-Rad, Hercules, CA, USA), protein bands were detected with Luminata Forte Western HRP substrate and quantified with a Bio-Rad chemiluminescence imaging system (Bio-Rad, Hercules, CA, USA). ImageJ 1.53 was used to measure the intensity of the band (Rawak Software, Inc., Stuttgart, Germany).

### 5.7. qRT-PCR

Trizol Reagent was used to extract RNA from colorectal tissues. RNA was reverse-transcribed with the HiScript^®^ II Q RT Supermix for quantitative real-time PCR. qRT-PCR was conducted using ChamQ Universal SYBR qRT-PCR Master Mix (Analytikjen, Germany) on a qTOWER3 Real-Time PCR System. -actin was utilized for mRNA expression normalization. Using the 2^−ΔΔCt^ method, the relative levels of mRNA were determined. Each experiment was repeated three times. Primer Premier 6.0 was utilized to design the primers shown in Table 1.

### 5.8. Statistical Analysis

Each experiment was performed a minimum of three times per modality. The data are presented as the mean and standard deviation, and statistical significance was determined when *p* < 0.05. A one-way ANOVA was conducted using SPSS version 22.0 (SPSS Inc., Chicago, IL, USA) to ascertain the statistical differences between the treatment groups.

## Figures and Tables

**Figure 1 toxins-15-00579-f001:**
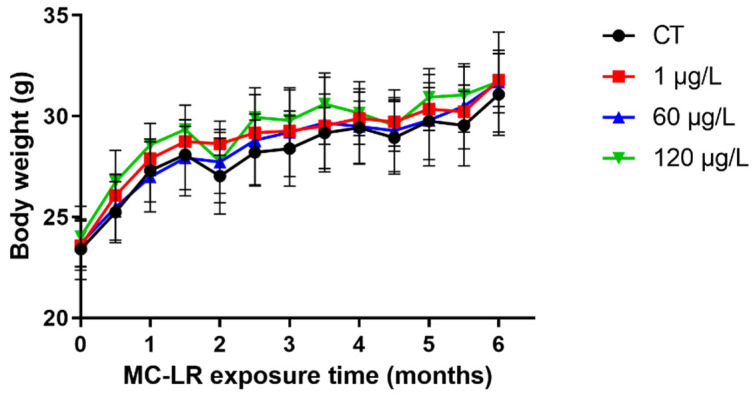
Effect of MC-LR on body weight. Data are presented as the mean ± SD.

**Figure 2 toxins-15-00579-f002:**
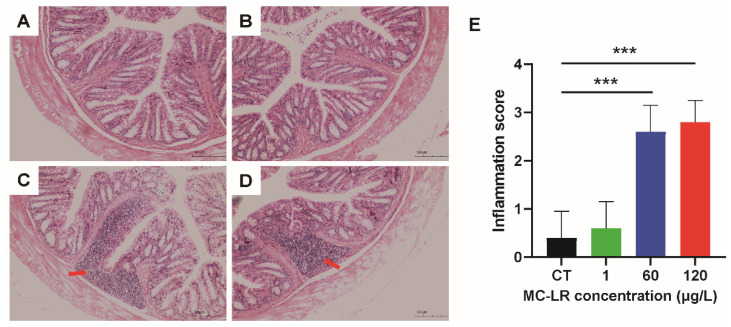
Effect of MC-LR on histopathological changes. (**A**) CT treatment group; (**B**) 1 µg/L MC-LR treatment group; (**C**) 60 µg/L MC-LR treatment group; (**D**) 120 µg/L MC-LR treatment group. (**E**) Inflammation score. The red arrow indicates lymphocyte infiltration. Bar = 50 μm means original magnification 50×. *** *p* < 0.001 compared with CT group.

**Figure 3 toxins-15-00579-f003:**
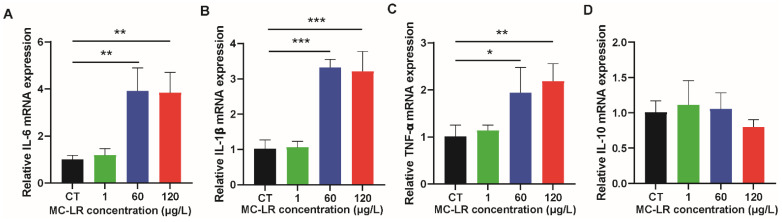
Effect of MC-LR on inflammatory factors. (**A**) The mRNA expression levels of IL-6; (**B**) the mRNA expression levels of IL-1β; (**C**) the mRNA expression levels of TNF-α; (**D**) the mRNA expression levels of IL-10. Data are presented as the mean ± SD; *n* = 5. * *p* < 0.05, ** *p* < 0.01, *** *p* < 0.001 compared with CT group.

**Figure 4 toxins-15-00579-f004:**
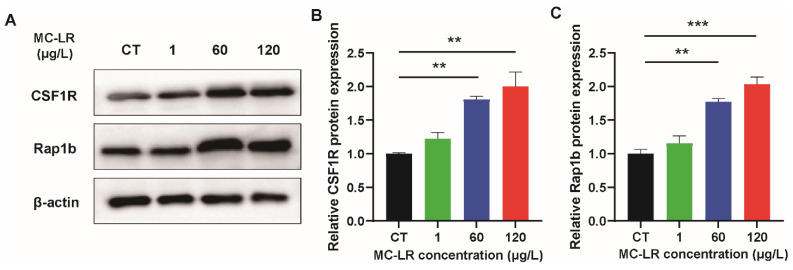
Effect of MC-LR on CSF1R/Rap1b signaling pathway. (**A**) WB analysis of proteins (CSF1R and Rap1b). (**B**,**C**) Relative quantitation of protein level normalized to β-actin. Data are presented as the mean ± SD; *n* = 5. ** *p* < 0.01, *** *p* < 0.001 compared with CT group.

**Figure 5 toxins-15-00579-f005:**
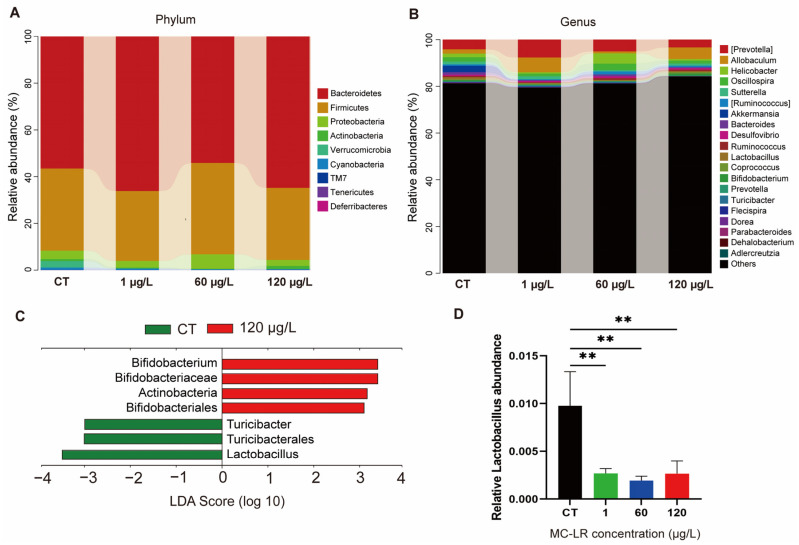
Effect of MC-LR on gut microbiome composition. (**A**) Representative phylum-level bacterial composition of the gut microbiome. (**B**) Representative genus-level bacterial composition of the gut microbiome. (**C**) The LDA score of the most differentially abundant taxa between CT and 120 µg/L MC-LR-treated groups as determined via LEfSe analysis. (**D**) Comparisons of the relative abundance of *Lactobacillus* between CT and MC-LR-treated groups. Data are presented as the mean ± SD; *n* = 5. ** *p* < 0.01 compared with CT group.

**Figure 6 toxins-15-00579-f006:**
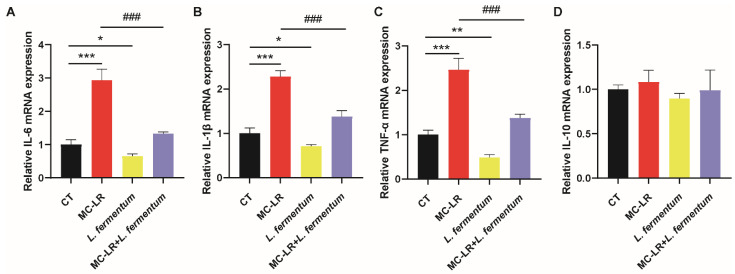
Effects of *L. fermentum* on CSF1R/Rap1b signaling pathway in MC-LR treatment cell model. (**A**) The mRNA expression levels of IL-6; (**B**) the mRNA expression levels of IL-1β; (**C**) the mRNA expression levels of TNF-α; (**D**) the mRNA expression levels of IL-10. Data are presented as the mean ± SD; *n* = 3. * *p* < 0.05, ** *p* < 0.01, *** *p* < 0.001 compared with CT group; ^###^ *p* < 0.001 compared with MC-LR treatment group.

**Figure 7 toxins-15-00579-f007:**
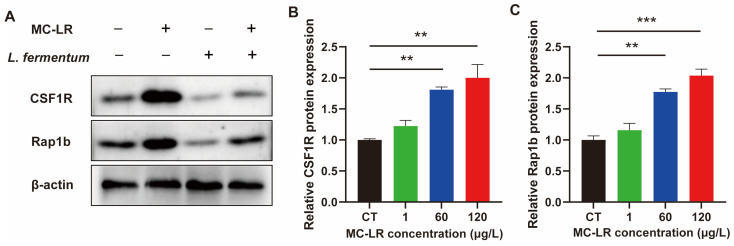
Effects of *L. fermentum* on inflammatory factors mRNA expression levels in MC-LR treatment cell model. (**A**) WB analysis of proteins (CSF1R and Rap1b). (**B**,**C**) Relative quantitation of protein level normalized to β-actin. Data are presented as the mean ± SD; *n* = 3. ** *p* < 0.01, *** *p* < 0.001 compared with CT group.

**Table 1 toxins-15-00579-t001:** Primer sequences for qRT-PCR.

Genes	Forward Primer (5′–3′)	Reverse Primer (5′–3′)
*M-IL-6*	CCACGGCCTTCCCTACTTC	TTGGGAGTGGTATCCTCTGTGA
*M-TNF-α*	CCCACGTCGTAGCAAACCA	ACAAGGTACAACCCATCGGC
*M-IL-1β*	GCACTACAGGCTCCGAGATGAA	GTCGTTGCTTGGTTCTCCTTGT
*M-IL-10*	AGAGCTGCGGACTGCCTTCA	ACCTGCTCCACTGCCTTGCT
*M-β-Actin*	TCAAGATCATTGCTCCTCCTGAG	ACATCTGCTGGAAGGTGGACA
*H-IL-6*	CCTTCGGTCCAGTTGCCTTCTC	AGAGGTGAGTGGCTGTCTGTGT
*H-TNF-α*	TGCTCCTCACCCACACCATCA	CCCAAAGTAGACCTGCCCAGAC
*H-IL-1β*	TCTGTACCTGTCCTGCGTGTTG	TCTGCTTGAGAGGTGCTGATGT
*H-IL-10*	TGTTGCCTGGTCCTCCTGACTG	CGCCTTGATGTCTGGGTCTTGG
*H-β-Actin*	GCACTCTTCCAGCCTTCCTTCC	CCGCCAGACAGCACTGTGTT

## Data Availability

Data will be made available on request.

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
