# Peer review of "Lactobacillus fermentum* Alleviates the Colorectal Inflammation Induced by Low-Dose Sub-Chronic Microcystin-LR Exposure"

_toxins, 2023, doi:10.3390/toxins15090579_

Round 1

Reviewer 1 Report

All the sections of the paper adequately describe the significance, importance and novelty of the paper. 

Results: The readers of the paper and the authors would benefit with HE staining images of the liver after exposure to the Lactobacillus culture to record any histopathological changes  observed.

Materials and Methods: The authors could also provide justification for the different concentrations of the toxin used.

Reviewer 2 Report

The article talks about the use of probiotic bacteria Lactobacillus fermentum to overcome the effects of MCLR toxicity.
- Names of bacteria should be in italics. Please check throughout the article.

- Please quantify the inflammatory foci/ infiltration of immune cells in Fig 2.

- Please elaborate in brief on the CSF1R/Rap1b pathway in the Introduction/ Discussion sections - Importance or outcome of the pathway and how MCLR is involved in activating it, etc.

- Fig 7 is mislabeled as Fig 6.

- Line 12 - .....protective mechanism of the mice resulted in tolerance. Spelling incomplete.

- Section 5.5 - what does the 200 mol/L dose in cells correspond to in your animal studies?
You mention that after seeding the cells and allowing them to adhere overnight, the cells were taken for treatment; 
-- Is there a serum free step that allows the cells to come to the same cell cycle stage before treatment? If yes, for how long?
-- Are the cells allowed to become confluent/grow in the well before treatment?
-- After adding the bacteria, are they allowed to aclimatize before adding the toxin?

- Is there literature that shows that treatment with Lactobacilli can restore the gut microbiota to normal levels?

- Do these bacteria possess the MCY operon that is instrumental in the breakdown of microcystin toxin? What is the mechanism of reduction in toxicity - are the bacteria taking up the toxin as energy source and breaking it down or does the toxin just adsorb to the bacterial cell wall thus, being less available for intestinal cells to cause an effect?

- Will there be animal studies with the probiotic as a therapeutic shown to reduce intestinal inflammation?

Minor edits addressed above.

Reviewer 3 Report

Authors have presented research aimed at evaluating the effect of probiotics regarding the intestinal toxicity induced by MC-LR using different techniques: histopathological analysis, metagenomic sequencing, western blotting, and qRT-PCR. For this purpose, a combination of in vitro and in vivo experiments was performed. The results indicate that MC-LR induced inflammation that can be mitigated by L. fermentum inhibiting the expression of CSF1R and Rap1b signalling pathway.

The methods of analysis appear appropriate, and the results do not appear to be over-interpreted. However, some information is unclear and discussion section should be widely checked. I would suggest a minor revision of the manuscript prior publication.

SPECIFIC COMMENTS FOR REVISION

-I have found some typographical errors in the document: missing commas, dots, italics… Especially in the discussion section (e.g., "...the mice resulted in a tolerate In addition...").

- The resolution of the figures should be drastically improved. It is difficult to read the graphs. Especially in figure 5 the font size is small and blurry. All figures should be changed.

- Insufficient information is given in introduction section about inflammatory pathways that later will be studied (CSF1R and Rap1b).

-  I would recommend rewriting the discussion section keeping the direct links of the discussion with the results, evaluating the possibility of transfer part of the information presented in discussion to introduction and keep the essentials points in discussion. (e.g., the information given in the second paragraph of the discussion).

- More information about experimental design and treatment should be explained:

            - Animals: Please explain the reason why mice (and especially male) were chosen. Why were 10 rats selected per group? Since the effects of MC-LR have been widely described, was there any way to reduce the number of animals used? Did authors follow any official guideline (e.g., OECD)?

- Cell lines: What was the survival rate of the NCM460 cells exposed to L.fermetum? Why did authors select one hour of incubation?
